# Ribozyme activity modulates the physical properties of RNA–peptide coacervates

Kristian Kyle Le Vay[1]*[†], Elia Salibi[1†], Basusree Ghosh[2], TY Dora Tang[2]*, Hannes Mutschler[1]*

[1]Department of Chemistry and Chemical Biology, TU Dortmund University, Dortmund, Germany; [2]Max-Planck Institute of Molecular Cell Biology and Genetics, Dresden, Germany

**Abstract** Condensed coacervate phases are now understood to be important features of modern cell biology, as well as valuable protocellular models in origin-of-life studies and synthetic biology. In each of these fields, the development of model systems with varied and tuneable material properties is of great importance for replicating properties of life. Here, we develop a ligase ribozyme system capable of concatenating short RNA fragments into long chains. Our results show that the formation of coacervate microdroplets with the ligase ribozyme and poly(L-lysine) enhances ribozyme rate and yield, which in turn increases the length of the anionic polymer component of the system and imparts specific physical properties to the droplets. Droplets containing active ribozyme sequences resist growth, do not wet or spread on unpassivated surfaces, and exhibit reduced transfer of RNA between droplets when compared to controls containing inactive sequences. These altered behaviours, which stem from RNA sequence and catalytic activity, constitute a specific phenotype and potential fitness advantage, opening the door to selection and evolution experiments based on a genotype–phenotype linkage.

## Editor's evaluation

Experimental models of simple cell-like compartments can help us to understand how biology operated early in its history. The authors convincingly show how the properties of coacervate droplets can be influenced by the activity of ribozymes inside them. This important result potentially provides a new route for biologists or chemists to establish cell mimics that support the evolution of biomolecules within.

## Introduction

Many biological biomolecular condensates are formed from RNA and proteins or peptides (*Roden and Gladfelter, 2021*; *Ukmar-Godec et al., 2019*), and coacervate phases formed by charge-mediated phase separation are part of the mechanism that drives the development of membraneless organelles in modern biology (*Hyman et al., 2014*; *Wang et al., 2021*). Condensed phases studied in the context of origin of life can also be formed from RNA and peptides. Catalytic nucleic acids, including ribozymes and DNAzymes, are central to 'Nucleic Acid World' hypotheses, where they act as both the medium of information storage and the catalyst for its replication in early life-like systems. Peptide–RNA condensation forms discrete protocellular compartments (*Abbas et al., 2021*; *Oparin, 1965*). RNA–peptide interactions are in many cases beneficial for the function of these early catalysts (*Frenkel-Pinter et al., 2020*; *Tagami et al., 2017*), and may promote the folding and oligomerisation of certain peptides (*Seal et al., 2022*). Perhaps surprisingly, the function of ribozymes and other nucleic acid enzymes in coacervate phases has only been recently established, with catalytic rate

*For correspondence:
kristian.levay@tu-dortmund.de
(KKLV);
tang@mpi-cbg.de (TYDT);
hannes.mutschler@tu-dortmund.de (HM)

†These authors contributed equally to this work

Competing interest: The authors declare that no competing interests exist.

being enhanced and enzyme function altered in some cases (*Drobot et al., 2018*; *Poudyal et al., 2019a*; ; *Poudyal et al., 2019b*; *Iglesias-Artola et al., 2022*). For example, coacervation with poly(L-lysine) shifts the reaction equilibrium of a minimal hairpin ribozyme from cleavage to ligation (*Le Vay et al., 2021*).

The ability of coacervate phases to strongly partition a wide range of molecular and macromolecular species (*Frankel et al., 2016*), and to support a variety of complex enzymatic processes (*Dora Tang et al., 2015*), makes them appealing proto- and artificial cell models in origin-of-life studies (*Ghosh et al., 2021a*), synthetic biology (*van Stevendaal et al., 2021*), and modern biology (*Yewdall et al., 2021*). Simple coacervates that form from oppositely charged polymers have been rigorously investigated for their ability to selectively partition biomolecules and host a variety of different chemistries and reactions in vitro (*Yewdall et al., 2021*; *Nakashima et al., 2019*). However, biological condensates formed by protein–protein or protein–nucleic acid interactions are dynamic systems, and their formation, dissolution, and physical properties are subject to spatiotemporal regulation (*Ismail et al., 2021*). Similarly, in order to realise a truly convincing model proto- or artificial cell, life-like behaviours such as growth, division, and other dynamic, responsive or non-equilibrium processes are essential. Several dynamic and responsive coacervate systems have been established, which are characterised by a phase change in response to environmental stimuli such as light exposure (*Martin et al., 2017*; *Huang et al., 2021*; *Kubota et al., 2022*), changes in temperature (*Lu et al., 2021*), or pH (*Love et al., 2020*; *Karoui et al., 2021*). Furthermore, non-equilibrium environments formed by gas bubbles inside heated rock pores have been shown to drive the growth, fusion, and division of otherwise inert coacervate microdroplets (*Ianeselli et al., 2022*).

Enzymatic processes that alter the properties of the coacervate components also affect coacervate properties and behaviour. In charge-based condensates, inducing phase change via enzymatic processes has been achieved by, for example, the conversion of ADP into charge-dense ATP (*Nakashima et al., 2018*), or the alteration of peptide charge state by phosphorylation (*Aumiller and Keating, 2016*). This has allowed the reversible generation of coacervate droplets by enzymatic networks. In addition, the polymerisation of UDP in $U_{20}$-spermine coacervates by polynucleotide phosphorylase has been shown to induce transient non-spherical coacervate morphologies (*Spoelstra et al., 2020*). All these systems depend on the action of proteinaceous enzymes, which presumably emerged later in molecular evolution, perhaps after the first protocells. However, droplets capable of dynamic change via the action of nucleic acid enzymes such as ribozymes have not been previously reported, partially because the catalytic repertoire of these catalysts is limited when compared to their proteinaceous counterparts (*Doudna and Lorsch, 2005*). Beyond environmental factors such as solution pH and salt concentration (*Li et al., 2018*), the physical properties and association behaviour of coacervate droplets are determined by factors such as component chain length (*Spruijt et al., 2010*) and charge density (*Neitzel et al., 2021*). Of these factors, we noted that polymer chain length could potentially be addressed by the action of a nucleolytic or ligase ribozyme. Thus, in a coacervate composed of RNA and a positively charged polymer, ribozyme-catalysed RNA cleavage or ligation might alter the physical properties or association behaviour of the system if a sufficiently large change in average RNA length were achieved. In particular, we hypothesised that the elongation of the RNA component could lead to the formation of a denser separated phase with altered physical properties (*Li et al., 2018*; *Liu et al., 2020*).

In this work, we demonstrate that coacervate microdroplets formed from a ligase ribozyme and lysine-based peptides display reciprocal and synergistic behaviour, in which coacervation enhances and modulates ribozyme activity, and ribozyme activity modulates droplet properties and therefore phenotype. We employ a modified ligase ribozyme that ligates short substrate strands into long concatemers, thus increasing the length of the polyanionic coacervate component. We find that coacervation enhances the rate of substrate ligation 50-fold and inhibits the formation of circular reaction products. In turn, the activity of the ribozyme imparts specific physical properties or phenotypes to the droplets it is contained within, which are not observed in droplets containing inactive ribozymes. These altered behaviours include the inhibition of droplet growth, surface wetting, and content exchange, which could provide fitness advantages under certain conditions. Connecting the sequence information of the ribozyme RNA to the physical properties of the resulting microdroplets is an example of a phenotype–genotype linkage, which is a fundamental requirement for the Darwinian evolution of protocellular systems (*Tawfik and Griffiths, 1998*).

# Results

To investigate the effect of ribozyme activity on coacervate behaviour, we initially sought to design a ribozyme system capable of increasing RNA chain length via concatenation. Although several examples of RNA ligase ribozymes have previously been reported (*Paul and Joyce, 2002*; *Ekland et al., 1995*; *Hayden et al., 2005*), these typically catalyse the ligation of a single junction, which results in only a modest overall increase in average RNA chain length. To achieve greater product lengths, we harnessed the catalytic core of the R3C ligase ribozyme ($E_R$), whose RNA ligation activity is based on 5′-triphosphate activated substrates (*Paul and Joyce, 2002*). This system has recently been shown to be active in the presence of poly(L-lysine) under certain conditions, and so was a promising starting point when considering compatibility with coacervate systems (*Iglesias-Artola et al., 2022*). We previously redesigned the ribozyme–substrate complex to iteratively produce long RNA concatemers from short oligonucleotides (*Matreux et al., 2021*). In our final design, the ribozyme ($E_L$) catalyses concatenation of a 31 nt substrate (*Figure 1a*). Screening of reaction conditions established that strong activity was observed at pH 8.6 in the presence of 10 mM $MgCl_2$ at a range of temperatures (30, 37, and 45°C) (*Figure 1—figure supplement 1*). The strongest activity was observed with a 1:1 monomer concentration ratio of substrate:ribozyme, while excess substrate inhibited the formation of longer length products at lower temperatures, most likely because the ribozyme has a higher probability of binding unligated substrates than already growing chains. In order to reduce hydrolytic degradation of RNA, we chose to maximise activity at 30°C with a 1:1 substrate:ribozyme ratio in all subsequent experiments.

The phase separation behaviour of the ribozyme system in the presence of poly(L-lysine) was initially investigated by titrating increasing amounts of $(Lys)_{19-72}$ into a fixed concentration of RNA (total monomer concentration = 1 mM) and measuring endpoint activity. The reaction products were separated by urea PAGE and stained using SYBR Gold to visualise all reaction components. The activity of the ribozyme was inhibited in the presence of excess peptide ($(Lys)_{19-72}$:RNA > 1) (*Figure 1b*), but yields of extended products at the endpoint of the reaction (2 hr) were enhanced above the solution reaction at lower ratios (lane profiles, *Figure 1b*). The reported concentration ratios are calculated from RNA and peptide monomer unit concentrations, and as such are also equivalent to charge ratios. The addition of $(Lys)_{19-72}$ led to a gradual increase in turbidity due to phase separation above the critical coacervation concentration of $CCC_{19-72} \approx 0.14:1$ $(Lys)_{19-72}$:RNA (*Figure 1c*). These experiments were repeated with a shorter peptide ($(Lys)_{5-24}$) (*Figure 1—figure supplement 2*), for which the onset of coacervation occurred at higher peptide:RNA ratios ($CCC_{5-24} \approx 0.93:1$ $(Lys)_{5-24}$:RNA), corroborating previous observations with poly(L-lysine) and the hairpin ribozyme (HPz) (*Le Vay et al., 2021*). In this case, the activity of the ribozyme was not inhibited in the presence of excess peptide, and again yields of extended products were enhanced above the solution condition (lane profiles, *Figure 1—figure supplement 2a*). For further experimentation, we selected specific peptide:RNA ratios of 0.75:1 $(Lys)_{19-72}$:RNA and 3:1 $(Lys)_{5-24}$:RNA, both of which supported coacervation and ribozyme catalysis. Different concentration ratios are required to produce similar droplets for each peptide due to their differing critical coacervation concentrations. Both selected concentration ratios corresponded to points immediately before the plateau in the respective peptide titration turbidity curves and allowed the formation of liquid coacervate droplets without suppression of ribozyme activity. (*Figure 1c* and *Figure 1—figure supplement 2b*). Fluorescence imaging of samples at these ratios confirmed the formation of phase-separated coacervate droplets that strongly partitioned the Cy5-labelled RNA substrate (*Figure 1d* and *Figure 1—figure supplement 2c*).

To investigate the kinetics of chain elongation and quantify the final product length, we conducted experiments using a 5′-fluorescently labelled substrate to visualise ligation products. At the chosen ratios, ligation rate and product length were greatly enhanced in the coacervate phase compared to solution (*Figure 1e and f* and *Figure 1—figure supplement 3*). The kinetic analyses show that the addition of either $(Lys)_n$ peptide resulted in an approximately 50-fold increase in the rate of concatenation and led to the formation of substrate chains with an average length approximately 30 nt greater than those produced in solution (*Figure 1e*). In solution, the kinetics of chain elongation were best approximated using a first order model ($k = 1.5 \times 10^{-2} \pm 1.0 \times 10^{-3}$ $min^{-1}$), and a final average product length (n) of 52.2 ± 0.5 nt was observed (*Figure 1e*). In the $(Lys)_{19-72}$ coacervate phase, the ribozyme kinetics were best described by a two-phase model ($k_{fast} = 7.7 \times 10^{-1} \pm 0.7 \times 10^{-1}$ $min^{-1}$, $k_{slow} = 6.6 \times 10^{-2} \pm 1.3 \times 10^{-2}$ $min^{-1}$), with a final product length of n = 77.8 ± 0.4 nt for $(Lys)_{19-72}$. Similar

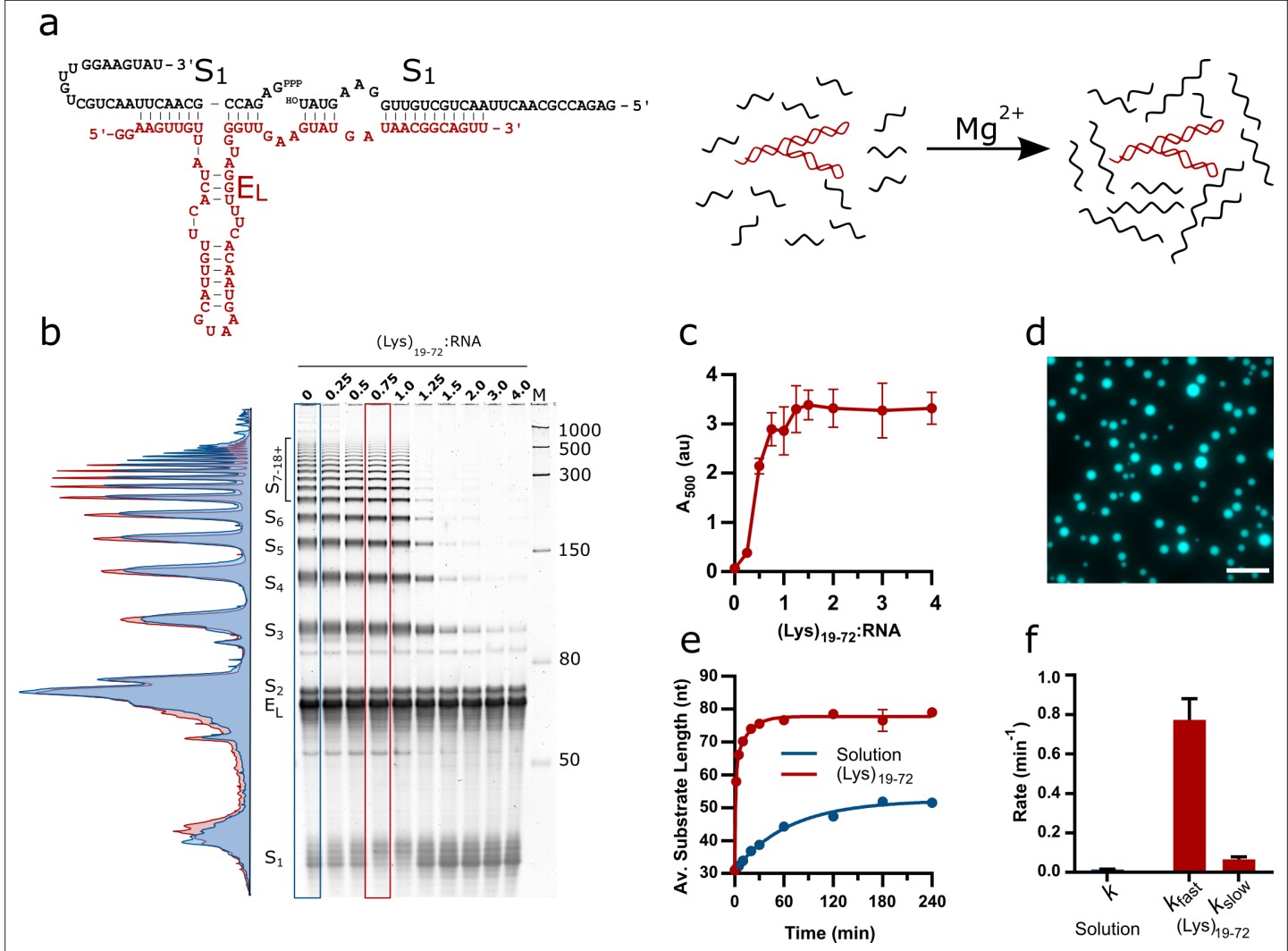

**Figure 1.** Design of the $E_L$ R3C ladder ribozyme system and activity in the presence of (Lys)$_{19-72}$ peptides. (**a**) The secondary structure of the ladder ribozyme and a schematic showing its function. The ribozyme is shown in red, whilst the substrate strands are shown in black. (**b**) A representative 8% urea PAGE stained with SYBR Gold showing the products of the R3C ladder system in solution and varying ratios of (Lys)$_{19-72}$ to R3C RNA (total monomer concentration = 1 mM, 10.5 μM substrate, 10.5 μM ribozyme) after a 2 hr reaction at 30°C in 50 mM Tris-HCl pH 8.6 and 10 mM MgCl$_2$. The integrated lane profiles of the solution and 0.75:1 (Lys)$_{19-72}$:RNA conditions are shown in blue and red, respectively (**c**) Variation in absorbance at 500 nm as a measure of coacervate formation upon addition of varying ratios of (Lys)$_{19-72}$ to the $E_L$ RNA after ligation for 3 hr at 30°C. Data points are an average of n = 3 independent replicates assembled from the same stock solutions. Error bars are standard deviations. (**d**) Example fluorescence microscopy image of (Lys)$_{19-72}$:RNA condensates at a ratio of 0.75:1 (Lys)$_{19-72}$:RNA, imaged using 10% Cy5-tagged substrate strand. Scale bar = 20 μm. (**e**) Kinetics of chain elongation in solution (blue, first-order model), and with 0.75:1 (Lys)$_{19-72}$:RNA (red, second-order model) at 30°C, pH 8.6, and 10 mM MgCl$_2$. A total RNA monomer concentration of 1 mM was achieved by combining 9.5 μM substrate, 1 μM Cy5-tagged substrate and 10.5 μM ribozyme. Data points are an average of n = 3 independent replicates assembled from the same stock solutions. Error bars are standard deviations. (**f**) Chain extension rate constants for the R3C ladder ribozyme in solution (blue, first-order model) and in the presence of 0.75:1 (Lys)$_{19-72}$:RNA (red, second-order model). Error bars are the standard errors for each parameter computed during non-linear regression. Equivalent data for condensates formed from the shorter (Lys)$_{5-24}$ peptide is shown in *Figure 1—figure supplement 2*.

The online version of this article includes the following source data and figure supplement(s) for figure 1:

**Source data 1.** Unedited and uncropped gel image for *Figure 1b*, and labelled image showing key bands and conditions.

**Source data 2.** Numerical turbidity data for *Figure 1c*.

**Source data 3.** Unprocessed and uncropped fluorescence microscope image for 0.75:1 (Lys)$_{19-72}$:RNA condensates (*Figure 1d*) imaged using 10% Cy5-tagged substrate strand.

**Source data 4.** Unedited and uncropped gel images for ribozyme kinetics (*Figure 1e*).

*Figure 1 continued on next page*

*Figure 1 continued*

**Figure supplement 1.** Characterisation of R3C ladder ribozyme activity at various temperatures and ribozyme:substrate ratios.

**Figure supplement 1—source data 1.** Unedited and uncropped gel image for *Figure 1—figure supplement 1*, and labelled image showing key bands and conditions.

**Figure supplement 2.** Activity of the $E_L$ R3C ladder ribozyme in the presence of short $(Lys)_{5-24}$ peptides.

**Figure supplement 2—source data 1.** Unedited and uncropped gel image for *Figure 1—figure supplement 2*, and labelled image showing key bands and conditions.

**Figure supplement 2—source data 2.** Numerical turbidity data.

**Figure supplement 2—source data 3.** Unprocessed and uncropped fluorescence microscope image for 3:1 $(Lys)_{5-24}$:RNA condensates imaged using 10% Cy5-tagged substrate strand.

**Figure supplement 2—source data 4.** Unedited and uncropped gel images for ribozyme kinetics (*Figure 1e*).

**Figure supplement 3.** Representative urea PAGE gels showing the activity of the $E_L$ R3C ladder ribozyme over time in solution and in the presence of short and long $(Lys)_n$ peptides.

**Figure supplement 3—source data 1.** Source data contains unedited and uncropped gel images used to estimate yields, as well as labelled images showing key bands and conditions and is identical to *Figure 1—source data 4*.

**Figure supplement 4.** The effect of RNA, $Mg^{2+}$, and PEG on the activity of the $E_L$ ribozyme.

**Figure supplement 4—source data 1.** Source data contains unedited and uncropped gel image, as well as labelled image showing the key bands and conditions.

**Figure supplement 5.** Formation of circular RNA products by the R3C ladder ribozyme.

**Figure supplement 5—source data 1.** Unedited and uncropped gel image, and labelled image showing key bands and conditions.

values were obtained for the short peptide (*Figure 1—figure supplement 2e*, Source data in *Figure 1—figure supplement 2—source data 4*). We have previously observed a coacervation-induced shift from monophasic to biphasic kinetic behaviour for the hammerhead ribozyme, albeit without an associated increase in rate (*Drobot et al., 2018*). High local RNA and $Mg^{2+}$ concentrations may lead to enhanced activity by promoting folding and increasing substrate and $Mg^{2+}$ cofactor abundance within the RNA–peptide coacervates (*Iglesias-Artola et al., 2022*; *Frankel et al., 2016*). To clarify origin of the enhancement reported here, we performed reactions at varying total RNA concentrations, varying $MgCl_2$ concentrations and in the presence of crowding agents (*Figure 1—figure supplement 4*). Briefly, doubling the current total RNA concentration increased product yields, but not to the level observed in the presence of poly(L-lysine), whilst halving the current total RNA concentration reduced product yields (*Figure 1—figure supplement 4a*). The addition of either PEG or increased $MgCl_2$ concentrations to the reaction mixture increased reaction yields relative to solution (*Figure 1—figure supplement 4b and c*), although again not to the same extent as phase separation with poly(L-lysine). We then tested the effect of combinations of PEG and $Mg^{2+}$ on activity (*Figure 1—figure supplement 4d*) and found that under certain conditions the presence of the crowding agent and additional $Mg^{2+}$ resulted in ligation yields similar to those observed in the RNA-peptide coacervates. These results suggest that multiple factors, including increased concentrations of both RNA and $Mg^{2+}$, are required for enhanced activity in the RNA-peptide coacervates.

In solution, the $E_L$ ribozyme produced both linear and circular concatenates, the latter of which are visible on urea PAGE gels as additional bands visible in between and above the regularly spaced linear

**Table 1.** Sequences of DNA oligomers used in this study.

| Name | Length | Sequence (5' to 3') |
|---|---|---|
| Substrate | 53 | ATACTTCCAACAGCAGTTAAGTTGCGGTCTC*TATAGTGAGTCGTATTAATTTC* |
| Active ribozyme | 87 | AACTGCCGTTATCATACTTCAACCCATCCAAAGTGTTACTTACGTAACAAGTGATAACAACTTCC*TATAGTGAGTCGTATTAATTTC* |
| Inactive ribozyme | 87 | AACTGCCGTTAAAATACAAAAACCCATCCACGCTGTTACGGACGTAACAGG TGATAACAACTTCC*TATAGTGAGTCGTATTAATTTC* |
| T7 promoter $P_{FWD}$ | 23 | GAAATTAATACGACTCACTATAG |

**Table 2.** Sequences of RNA oligomers used in this study.

| Name | Length | 5' end | Sequence (5' to 3') | Source |
|---|---|---|---|---|
| Substrate | 31 | Triphosphate | GAGACCGCAACUUAACUGCUGUUGGAAGUAU | IVT |
| Fluorescent sub | 31 | FAM or Cy5 | GAGACCGCAACUUAACUGCUGUUGGAAGUAU | IDT |
| Active ribozyme | 65 | - | GGAAGUUGUUAUCACUUGUUACGUAAGUAACA CUUUGGAUGGGUUGAAGUAUGAUAACGGCAGUU | IVT/IDT |
| Inactive ribozyme | 65 | - | GGAAGUUGUUAUCAC**C**UGUUACGU**CC**GUAACA **GCG**UGGAUGGGUUU**UUU**GUAU**UU**UAACGGCAGUU | IVT/IDT |

Red bases represent inactivating mutations.
IVT: in vitro transcription.

ladder of products due to the different mobility of linear and circular products through the gel matrix. The presence of circular products was confirmed by treatment of reacted RNA with RNase R, a 3′ to 5′ exoribonuclease that only digests linear strands (*Figure 1—figure supplement 5*). The presence of the long peptide (0.75:1 (Lys)$_{19-72}$:RNA) suppressed circularisation altogether, whilst the short peptide (3:1 (Lys)$_{5-24}$:RNA) reduced the formation of circular products.

Having demonstrated that the E$_L$ ribozyme is capable of substrate concatenation in both solution and in (Lys)$_n$ coacervate droplets, and that its activity is enhanced within the coacervate environment, we aimed to determine whether the elongation of the RNA component would lead to differences in droplet properties, thus altering phenotype. To identify changes due solely to chain concatenation, we developed an inactive mutant of the ribozyme by introducing several point mutations ('Methods', *Table 1* and *Table 2*), without altering the substrate binding region. Ribozyme activity assays revealed that the mutant was completely inactive in solution and in the presence of both (Lys)$_n$ peptides at the previously specified charge ratios (*Figure 2—figure supplement 1*). Populations of droplets containing either the active or inactive ribozyme were loaded into a passivated well plate and imaged over the course of 24 hr. In these experiments, (Lys)$_n$ was added to the RNA reaction mixture immediately, so that all ligation activity would occur in the presence of the peptide. In all cases, phase-separated droplets were observed which persisted over the course of the experiment (*Figure 2a and c* and *Figure 2—figure supplement 2a and c*). Whilst many droplets remained static, some coarsening to form larger droplets was observed, particularly in the droplet populations containing the inactive ribozyme. To fully quantify these observations, a segmentation algorithm was used to identify droplets and thereby measure their size and population density (*Stringer et al., 2021*). Droplets which contained active ribozyme maintained a constant average area over the 24 hr period, whilst droplets containing inactive ribozyme grew, in some cases reaching a final area more than twice that of the active system (*Figure 2e* and *Figure 2—figure supplements 2 and 3*). As the average droplet area increased for the inactive droplets, there was a concomitant decrease in droplet population density (the average number of droplets per 100 μm$^2$), indicating that the growth was at least, in part, due to coalescence events between the droplets (*Figure 2f*). A minimal decrease in droplet population density was observed for active droplets, suggesting that lack of coalescence prevented growth. Similar behaviour was observed for both the long and short peptide (*Figure 2— figure supplements 2 and 3*), although for the short peptide system, the number of both active and inactive droplets decreased similarly over the course of the experiment, indicating that coalescence was not completely supressed in the active droplets. Intriguingly, some particles formed from the short peptide and active ribozyme system initially adopted a non-spherical morphology and relaxed to form round droplets over the course of the experiment (*Figure 2—figure supplement 4*), similar to morphological changes induced by the extension of UDP in spermine-based coacervates (*Spoelstra et al., 2020*). This indicates that the material properties of the droplets change over time.

Whilst surface passivation is used as standard in the imaging of coacervate systems to prevent wetting and adhesion effects, surface interactions can affect droplet formation and morphology, and therefore may also influence the observed protocellular phenotype. Consequently, we repeated the previous experiment on an unpassivated polystyrene surface (Greiner μclear microplate, medium binding). The coacervate droplets containing the active ribozyme behaved as previously observed, with discrete round morphologies (*Figure 2b*) and little change in both average droplet area and

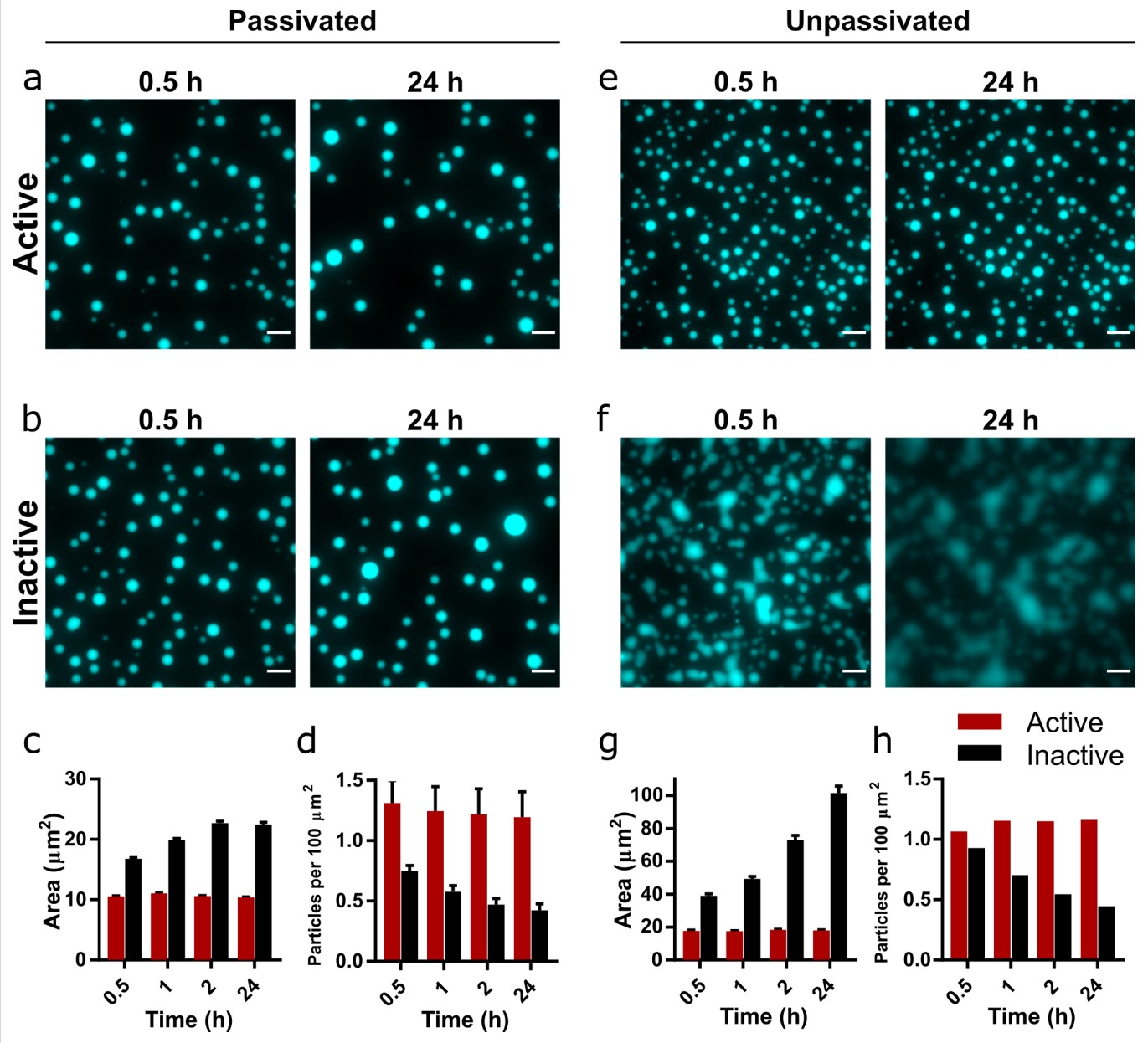

**Figure 2.** Droplet morphology over time in active and inactive coacervate systems formed from R3C RNA and (Lys)$_{19-72}$. Representative images of coacervate droplets prepared with active (**a, e**) or inactive ribozyme (**b, f**) and 0.75:1 (Lys)$_{19-72}$:RNA in passivated (**a, b**) and unpassivated (**e, f,**) environments. Scale bars = 10 µm. Average particle areas and population density over time for the passivated environment are shown in (**c**) and (**d**), respectively. Equivalent data for the unpassivated environment are shown in (**g**) and (**h**). All experiments were performed with 1 mM total RNA monomer concentration, a 0.75:1 ratio of (Lys)$_{19-72}$:RNA monomers, and at 30°C, pH 8.6, and 10 mM MgCl$_2$. The RNA reaction mixture contained 10% Cy5-labelled substrate for fluorescence imaging. Particles were measured from at least nine separate images, except for unpassivated samples for which a single image was captured. Error bars are standard errors. Data from active populations are shown in red, while data from inactive populations are shown in black. The number of droplets analysed was n=1702 - 5272 for the passivated environment and n = 199 - 518 for the unpassivated environment. Droplet areas and particle counts were measured using the CellPose segmentation algorithm (***Stringer et al., 2021***). Data for condensates formed from the shorter (Lys)$_{5-24}$ peptide are shown in ***Figure 2—figure supplement 2***.

The online version of this article includes the following source data and figure supplement(s) for figure 2:

**Source data 1.** Unprocessed and uncropped fluorescence microscope image for 0.75:1 (Lys)$_{19-72}$:RNA condensates on passivated surfaces imaged using 10% Cy5-tagged substrate strand.

**Source data 2.** Unprocessed and uncropped fluorescence microscope image for 0.75:1 (Lys)$_{19-72}$:RNA condensates on unpassivated surfaces imaged using 10% Cy5-tagged substrate strand.

*Figure 2 continued on next page*

*Figure 2 continued*

**Figure supplement 1.** Comparison of active and inactive ribozyme variants.

**Figure supplement 1—source data 1.** Unedited and uncropped gel image, and labelled image showing key bands and conditions.

**Figure supplement 2.** Development of droplet morphology over time in active and inactive coacervate systems formed from R3C RNA and $(Lys)_{5-24}$ peptide.

**Figure supplement 2—source data 1.** Unprocessed and uncropped fluorescence microscope image for 3:1 $(Lys)_{5-24}$:RNA condensates on passivated surfaces imaged using 10% Cy5-tagged substrate strand.

**Figure supplement 2—source data 2.** Unprocessed and uncropped fluorescence microscope image for 3:1 $(Lys)_{5-24}$:RNA condensates on unpassivated surfaces imaged using 10% Cy5-tagged substrate strand.

**Figure supplement 3.** Frequency histograms area of Cy5-labelled coacervate droplet area over time.

**Figure supplement 4.** Circularity of Cy5-labelled coacervate droplets over time.

**Figure supplement 5.** Comparison between condensates formed from pre-reacted $E_L$ ribozyme and substrate and $(Lys)_n$ at 1 hr and 24 hr after mixing.

**Figure supplement 5—source data 1.** Unprocessed and uncropped fluorescence microscope image for 0.75:1 $(Lys)_{19-72}$: pre-reacted RNA condensates imaged using 10% FAM-tagged substrate strand.

**Figure supplement 5—source data 2.** Unprocessed and uncropped fluorescence microscope image for 3:1 $(Lys)_{5-24}$: pre-reacted RNA condensates imaged using 10% FAM-tagged substrate strand.

population density over the course of the experiment (*Figure 2g and h*). These droplets exhibited greater average areas than those in the passivated environment, likely due to wetting onto the surface. In contrast, droplets containing the inactive ribozyme wet the surface and rapidly spread, eventually merging to form a film of the condensed coacervate phase on the bottom of the well (*Figure 2d*). The measured average particle area therefore increased greatly over the course of the experiment, whilst the number of particles decreased (*Figure 2g and h*).

These differences in behaviour indicate that the generation of longer RNA within the coacervate environment imparts altered physical properties on the droplets, despite maintaining a spherical morphology typical of a liquid system. To further investigate this phenomenon, we added the $(Lys)_{5-24}$ and $(Lys)_{19-72}$ peptides to pre-reacted RNA mixtures. Here, we observed that mixtures containing the active ribozyme and therefore pre-concatenated RNA initially formed non-spherical gel-like condensates with both peptides, which relaxed to form spherical droplets over the course of 24 hr, whilst mixtures with inactive ribozyme yielded spherical droplets from the outset (*Figure 2—figure supplement 5*). This suggests that large morphological differences between active and inactive systems are not observed when peptide is added to ribozyme and substrate mixtures without pre-reaction because the average RNA length is initially identical. As the reaction proceeds in the active systems, a transition to a more viscous or gel-like state may occur whilst maintaining the initially formed spherical morphology.

Given the effect of RNA concatenation on the coacervate droplets, we asked whether the activity of the $E_L$ ribozyme could affect the interactions between different populations of droplets. Using the previously established conditions and concentration ratios, we mixed populations of droplets containing either a Cy5- or FAM-tagged substrate (10% total substrate concentration) and monitored mixing and content exchange over the course of 24 hr. These effects can be quantified by the calculation of a Pearson correlation coefficient (PCC), which measures the correlation of pixel intensities between the two fluorescence channels (*Dunn et al., 2011*). Two coefficients were calculated: $PCC_{droplet}$, which measures the degree to which fluorophores mix within individual fused droplets, and $PCC_{pop}$, which is calculated on a population level and measures the degree of mixing between the two populations. Positive values indicate spatial colocalisation of fluorophores, whilst negative values indicate spatial separation of fluorophores and therefore the presence of discrete regions or populations.

Shortly following mixing, discrete populations of Cy5- and FAM-labelled droplets are clearly visible (*Figure 3a and b* and *Figure 3—figure supplement 1*). Fused and unevenly mixed droplets containing both fluorophores are visible, providing evidence that coalescence is the mechanism of droplet growth in this system. After 24 hr, visual inspection of the images reveals that discrete populations of Cy5- and FAM-labelled droplets only persist in droplets containing the active ribozyme (*Figure 3a* and *Figure 3—figure supplement 1a*). In the inactive droplets, both fluorophores appear evenly distributed throughout the population (*Figure 3b* and *Figure 3—figure supplement 1b*). In all cases, the $PCC_{droplet}$ increased over time, tending towards unity, indicating that labelled RNA was

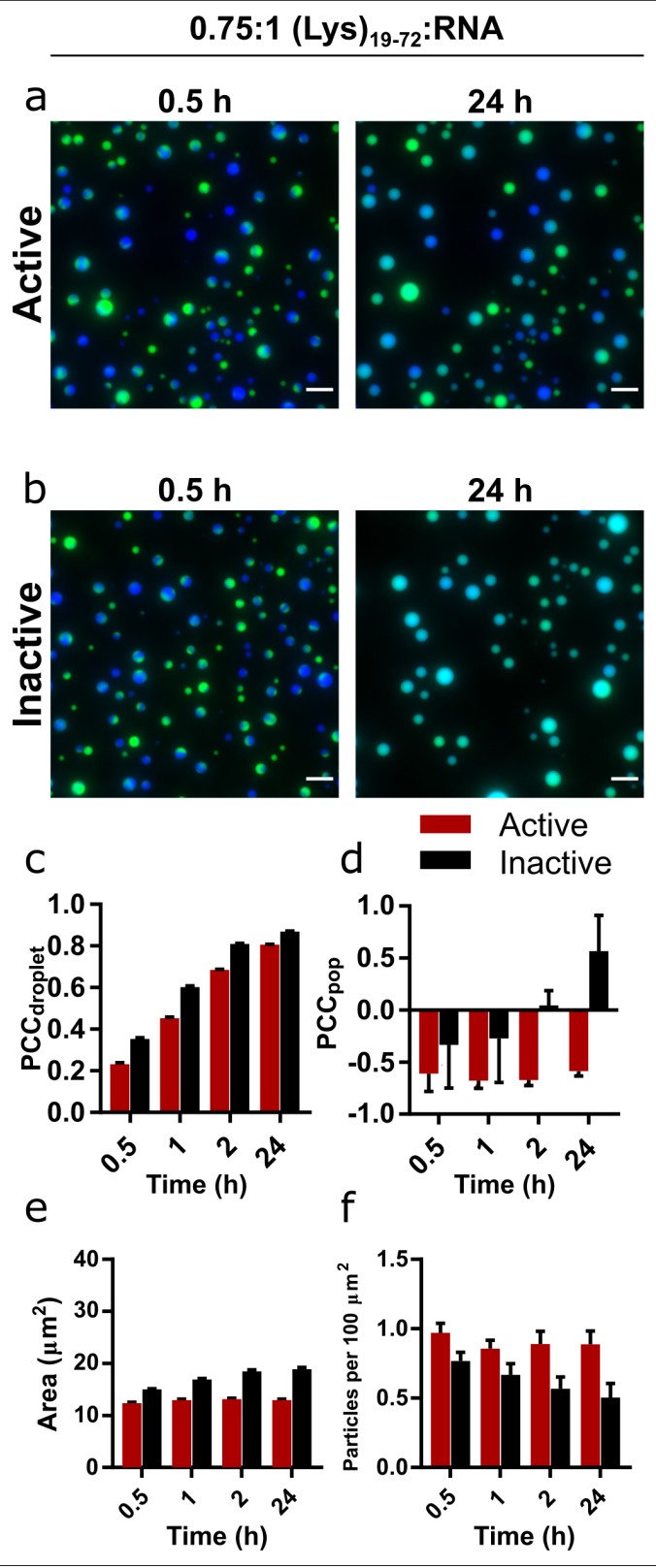

**Figure 3.** Mixing and content exchange between populations of active or inactive coacervate systems formed from R3C RNA and (Lys)$_{19-72}$ peptide. (**a, b**) Example images of mixtures of orthogonally labelled coacervate droplets prepared with 0.75:1 (Lys)$_{19-72}$:RNA containing either active (**a**) or inactive (**b**) ribozyme. Each population in the set contained either 10% FAM- or 10% Cy5-tagged substrate (green and blue, respectively). The two populations

*Figure 3 continued on next page*

*Figure 3 continued*

in each set were mixed shortly after preparation and then imaged over 24 hr in a passivated environment. All experiments were performed at 30°C, pH 8.6, and 10 mM $MgCl_2$ with a 1 mM total RNA monomer concentration and a 0.75:1 ratio of $(Lys)_{19-72}$:RNA. The colocalisation of the two fluorophores within single droplets is measured by the droplet Pearson coefficient ($PCC_{droplet}$) (**c**), whilst the colocalisation of fluorophores in the overall population of droplets is measured by the population Pearson coefficient ($PCC_{pop}$) (**d**). The average particle area and number of particles per unit area over time are shown in (**e**) and (**f**) respectively. Particles were measured from at least six separate images. Data from active populations are shown in red, while data from inactive populations are shown in black. The number of droplets analysed was n = 1350 - 3174. Error bars are standard errors. Scale bars = 10 μm. Data for condensates formed from the shorter $(Lys)_{5-24}$ peptide are shown in *Figure 3—figure supplement 1*.

The online version of this article includes the following source data and figure supplement(s) for figure 3:

**Source data 1.** Unprocessed and uncropped fluorescence microscope images of mixed populations of 0.75:1 $(Lys)_{19-72}$:RNA condensates containing active ribozyme imaged in both FAM and Cy5 channels.

**Source data 2.** Unprocessed and uncropped fluorescence microscope images of mixed populations of 0.75:1 $(Lys)_{19-72}$:RNA condensates containing inactive ribozyme imaged in both FAM and Cy5 channels.

**Figure supplement 1.** Mixing and content exchange between populations of active or inactive coacervate systems formed from R3C RNA and the $(Lys)_{5-24}$ peptide.

**Figure supplement 1—source data 1.** Unprocessed and uncropped fluorescence microscope images of mixed populations of 3:1 $(Lys)_{5-24}$:RNA condensates containing active ribozyme imaged in both FAM and Cy5 channels.

**Figure supplement 1—source data 2.** Unprocessed and uncropped fluorescence microscope images of mixed populations of 3:1 $(Lys)_{5-24}$:RNA condensates containing inactive ribozyme imaged in both FAM and Cy5 channels.

**Figure supplement 2.** Comparison of mixing of labelled RNA in individual fusion droplets over time measured using the droplet Pearson correlation coefficient ($PCC_{droplet}$).

**Figure supplement 2—source data 1.** Numerical data for all plots displayed in *Figure 3—figure supplements 2 and 3*.

**Figure supplement 3.** Intensity scatter plots showing the colocalisation of FAM- and Cy5-labelled RNA in mixed droplet populations.

---

able to equilibrate within the droplets via diffusion (*Figure 3c* and *Figure 3—figure supplement 1c*). However, it is notable that the rate and magnitude of increase in $PCC_{droplet}$ was greater in droplets containing the inactive ribozyme (*Figure 3—figure supplement 2*), indicating greater RNA mobility. For active and inactive $(Lys)_{19-72}$ droplets, $PCC_{pop}$ was initially negative, implying discrete and distinct populations (*Figure 3d*). Over the course of the experiment, the active system maintained this separation, whilst the $PCC_{pop}$ of the inactive system tended towards unity. These trends are further confirmed by scatter plots of normalised FAM and Cy5 fluorescence intensity in individual droplets (*Figure 3—figure supplement 3*), which show the clearly separated populations of active systems and mixing in inactive systems. As observed in our previous experiments, mixtures of the active ribozyme and peptide produced smaller droplets which did not grow over the course of the experiment compared to inactive populations, suggesting that droplets with inactive ribozyme are more prone to coalesce and therefore mixing than droplets containing the active ribozyme (*Figure 3e and f*).

A similar yet weaker trend was observed with short $(Lys)_{5-24}$ peptide droplets, which exhibited greater content exchange overall between both active and inactive droplets (*Figure 3—figure supplements 1–3*). Here, the presence of the active ribozyme slowed the rate of coalescence and therefore content exchange rather than completely suppressing it, but still allowed active populations to maintain a limited degree of identity over the course of the experiment.

## Discussion

Our original goal was to develop a ribozyme capable of RNA chaining that could significantly increase the average RNA length in a coacervate system, and thereby allow us to study the influence of such a reaction on the material properties of the coacervates. Although ribozyme systems that concatenate a short substrate have been previously reported, these do so via a reversible cyclic-phosphate-mediated mechanism, and as such the concentration of extended products decreases rapidly as product length increases (*Mutschler et al., 2015*). The R3C ligase ribozyme was thus an attractive starting point for the development of our system as the 5'-triphosphate activated reaction it catalyses is quasi-irreversible,

and the ribozyme is already optimised for high rate and yield (**Paul and Joyce, 2002**). Indeed, the $E_L$ ribozyme was able to generate RNA strands of >500 nt in length, including circular species, and this activity was enhanced in terms of both rate and yield in $(Lys)_n$ coacervates. Although the observation of circular products in concatenation reactions has been previously reported in ribozyme systems (**Jeancolas et al., 2021**), the inhibition of this behaviour by coacervation is notable. Whilst the mechanism of this inhibition is unknown, the absence of circular products suggests that the ends of a substrate strand do not meet, hinting at a decrease in RNA mobility, increased chain stiffness, or reduced release of ligated substrate from the ribozyme within the condensed phase compared to the bulk solution. The suppression of circularisation should lead to higher overall RNA lengths as the concentration of RNA ends available for ligation is not decreased over the course of the reaction. It is challenging to quantify this difference under the current experimental setup, as measurement of the average chain lengths requires the use of a 5' fluorescent tag (as in the kinetics experiment), which itself prevents formation of circular products due to the blocked 5' end. A potential solution to overcome this limitation would be to use a substrate with an internal fluorophore (assuming it does not affect ligation activity) or a radiolabel.

We have previously reported that the activity of the hairpin ribozyme (HPz) is greatly enhanced in condensed phases comprised of catalytic RNA and $(Lys)_n$ peptides (**Le Vay et al., 2021**). The increase in turbidity observed on titrating either $(Lys)_{19-72}$ or $(Lys)_{5-24}$ into a fixed concentration of R3C ribozyme and substrate was similar to that observed for HPz, although in this study the resultant condensed phase appeared liquid rather than gel-like. This variation in condensed phase morphology may be due to the influence of RNA structure, sequence, and hybridisation state on condensation (**Vieregg et al., 2018**; **Boeynaems et al., 2019**), with a higher degree of single-stranded or loop structures in the $E_L$ ribozyme system. The inhibition of ribozyme activity at excess $(Lys)_{19-72}$:RNA ratios was comparable to that observed in our previous study (**Le Vay et al., 2021**) and is attributed to a peptide length-dependent melting or misfolding of native nucleic acid tertiary structures at high concentrations of longer peptides (**Cakmak et al., 2020**; **Nott et al., 2016**). A previous study into the function of the R3C replicase ribozyme in RNA–peptide coacervates reported a similar trend: catalytic activity was inhibited at excess $(Lys)_n$:RNA ratios for a range of peptides (n = 7, 8, 9, 10, 18, and 19–72) (**Iglesias-Artola et al., 2022**). The present study extends the peptide concentration range in which the R3C ribozyme functions, with robust catalytic activity at ratios as high as 4:1 $(Lys)_{5-24}$:RNA. A possible explanation is that commercially available $(Lys)_n$ deviates from the manufacturer's stated size range, with the $(Lys)_{5-24}$ used here being predominantly comprised of oligomers between n = 3 to n = 9 in length (**Le Vay et al., 2021**). These short fragments may interact only weakly with the RNA, effectively reducing the concentration of peptide oligomers that can form condensates. In general, it should be noted that the effect of phase separation on ribozyme activity in a given system is highly specific to the identity of the ribozyme and peptide, their relative concentrations, and environmental conditions such as buffer, magnesium concentration, and temperature.

The difference in growth behaviour between active and inactive droplets is a striking demonstration of how the presence of a catalytic species can modulate the physical properties and behaviour of a model protocellular system, in this case by reducing coalescence. The coarsening of liquid coacervate phases is not inevitable: growth can be also be suppressed by active chemical processes hosted within droplets that produce the polymer components (**Zwicker et al., 2015**), although in our experiments the action of the ribozyme does not increase the concentration of RNA or peptide in the system. The formation of kinetically trapped states can also allow the persistence of static droplet populations over time (**Ranganathan and Shakhnovich, 2020**). However, increasing polymer lengths in a coacervate system results in stronger cooperative electrostatic interactions between polymer components (**Li et al., 2018**), and has been shown to affect physical parameters such as droplet water content, critical salt concentration, and condensed phase polymer concentration (**Spruijt et al., 2010**), leading to denser droplets and a more depleted dilute phase. Simulations have shown that increasing RNA length modulates both the material and interfacial properties of RNA–peptide condensates, in particular increasing density and surface tension (**Laghmach et al., 2022**). RNA length also has been shown to modulate viscosity in biomolecular condensates (**Tejedor et al., 2021**), which in turn determines the droplet fusion dynamics (**Ghosh et al., 2021b**). Thus, increasing overall RNA length by R3C substrate concatenation may inhibit growth via coalescence by increasing condensate density and viscosity, and reduce wetting affects through the alteration of surface tension. Whilst RNA sequence and secondary

structure have also been shown to influence the viscoelastic behaviour of RNA peptide condensates (*Alshareedah et al., 2021*), the active and inactive ribozyme systems used in the present study differ by only few nucleotides.

Despite the spherical morphology of the RNA–peptide condensates reported here, the inhibition of growth and wetting may also be due to a phase transition to a gel state. The formation of non-liquid RNA–peptide condensates has been previously reported for poly-rA:rU-peptide mixtures, where RNA base pairing leads to the formation of a kinetically arrested gel-like solid (*Boeynaems et al., 2019*), as well as in our previous work (*Le Vay et al., 2021*). These gel condensates melt to form spherical particles upon thermal denaturation and annealing, which disrupts the networked structures formed from complementary RNA strands and peptides. In our R3C system, complementary base-pairing interactions between substrate and ribozyme should also permit the formation of large, non-covalently assembled networked structures due to the repeating nature of the substrate. Ribozyme activity is expected to increase the stability of these structures as substrate ligation increases the free energy of association between ribozyme and substrate. We hypothesise that in the inactive system, and before significant concatenation occurs in the active system, RNA–RNA interactions are sufficiently weak to allow the formation of liquid coacervate droplets. In the active system, liquid droplets initially form because the timescale of coalescence is faster than that of RNA ligation. However, as ligation proceeds the stability of networked assemblies of ribozyme and substrate increases, resulting in a transition from a liquid to either a highly viscous liquid state or solid gel state that is both unable to grow and or wet the unpassivated surface on the timescale of the experiment. This hypothesis is supported by our observation that mixtures of pre-concatenated RNA and (Lys)$_n$ initially form non-spherical gel-like aggregates, which then relax to form spherical droplets over the course of 24 hr.

The reduction of mixing between populations of droplets containing orthogonally labelled RNA likely results from the changes in material properties and growth behaviour caused by RNA concatenation. The coalescence of the coacervate droplets observed in all inactive systems is likely to be the primary mechanism of mixing in this scenario. Indeed, greater droplet growth was observed in the two-population system than for single-population experiments, which may be due to the necessity of pipetting each droplet population into the sample environment sequentially, thus increasing mechanical agitation and contacts between droplets. All populations at the initial time point contain fusion droplets, which contain separate and unmixed areas of each fluorescent RNA. Notably, fusion droplets in all systems appear evenly mixed after 24 hr, suggesting that significant equilibration occurs before any transition to a highly viscous liquid or solid gel. Mixing may also occur by diffusive transfer of RNA between droplets. As the fluorescently labelled substrate is ligated onto the growing concatemer chain, the diffusion coefficient of the labelled material is expected to decrease over the course of the reaction as its effective length increases, which would in turn slow diffusive transfer in the active system.

Taken together, the results reported here describe a range of altered droplet behaviours stemming from ribozyme-driven changes in coacervate physical properties. These physical changes may impact other processes relevant to early compartmentalised replicators and protocells, such as resistance to molecular parasites (*Koonin et al., 2017*) and metabolite exchange (*Smokers et al., 2022*), and thus are exciting avenues for further study. The ability of a population of protocells to resist passive coarsening, surface wetting, and content exchange can already be considered a fitness advantage (*Nakashima et al., 2021*), and the emergence of fitness differences from varying protocellular compositions and phenotypes is a key step on the path towards Darwinian evolution with suitable selective pressures (*Adamala and Szostak, 2013*). Furthermore, in this system the phenotypic difference originates from the ribozyme-driven concatenation reaction, which increases the length of the anionic coacervate component. The sequence of the ribozyme in effect comprises the genotype of our protocellular system, with different variants (e.g. active or inactive sequences) producing different protocellular phenotypes. The realisation of a phenotype–genotype linkage is essential for true open-ended evolution, otherwise phenotypes with obvious fitness advantages have no possibility of being inherited by future generations. We envision that recent methodological developments in RNA sequencing from single coacervate microdroplets will enable future selection experiments on populations of protocells with varying RNA genotypes and varying degrees of fitness with respect to environmental pressures such as heat or salt concentration (*Wollny et al., 2022*).

The reciprocal modulation of ribozyme activity and coacervate properties reported here furthers the argument for the early coevolution of RNA and peptides (*Frenkel-Pinter et al., 2020*; *Tagami et al., 2017*; *Seal et al., 2022*; *Ghosh et al., 2021a*; *Le Vay and Mutschler, 2019*). However, further development is required to move towards more realistic model systems. A key limitation is that the droplets presented in this work contain a large proportion or pure ribozyme and substrate, which would presumably be less abundant in a more plausible scenario. The R3C ligase is a highly optimised system developed by in vitro selection and offers little potential for further enhancement of catalytic activity, but similar material changes with less ribozyme might be achieved by optimising the substrate binding arms and conditions for multiturnover reaction, or tuning substrate length to produce greater overall changes with fewer ligations. We have previously demonstrated substrate strand release in hairpin ribozyme-$(Lys)_n$ condensates (*Le Vay et al., 2021*), so it may be possible to engineer similar behaviour in the R3C system, perhaps at the cost of reaction rate. In addition, the simple lysine polypeptides used here are a very basic model system, but additional variation in droplet properties may be achieved in future by varying peptide sequence. Droplet populations formed from different peptides could exhibit varying degrees of ribozyme enhancement or fundamentally different physical properties that allow selection based on peptide identity as well as RNA sequence and activity. Although the magnitude of behavioural change was greater for coacervate populations formed with the longer peptide, clear phenotypic differences were nonetheless also present in the $(Lys)_{5-24}$ system. This short peptide, predominantly composed of n = 3–9 residue oligomers, is of a length that could be produced by prebiotically plausible processes such as wet–dry cycling (*Forsythe et al., 2015*; *Rodriguez-Garcia et al., 2015*).

## Materials

Trizma base (Tris; Thermo Fisher Scientific, Waltham, MA), sodium hexametaphosphate ($(NaPO_3)_6$, 611.77 g/mol; Sigma-Aldrich, St. Louis, MI), formamide ($CH_3NO$, 45.04 g/mol; Sigma-Aldrich), ethylenediaminetetracetic acid disodium salt dihydrate (EDTA, $C_{10}H_{14}N_2Na_2O_8 \cdot 2H_2O$, 372.24 g/mol; Sigma-Aldrich), magnesium chloride hexahydrate ($MgCl_2 \cdot 6H_2O$, 203.30 g/mol; (Merck, Darmstadt, Germany)), sodium hydroxide (NaOH, 39.997 g/mol; VWR, Radnor, PA), sodium chloride (NaCl, 58.44 g/mol; Sigma-Aldrich), urea ($CH_4N_2O$, 60.06 g/mol; Carl Roth, Karlsruhe, Germany), hydrochloric acid (HCl, 37%, 36.46 g/mol) (VWR), boric acid ($H_3BO_3$, 61.83 g/mol; Merck), ammonium persulfate (APS, $(NH_4)_2S_2O_8$, 228.20 g/mol) (VWR), acrylamide (19:1 bisacrylamide; Thermo Fisher Scientific), tetramethylethylendiamine (TEMED, $C_6H_{16}N_2$, 116.21 g/mol; Carl Roth), SYBR gold stain (Thermo Fisher Scientific), RNA oligomer length standard (low-range ssRNA ladder) (NEBm Ipswich, MA).

All peptides were purchased from Sigma-Aldrich and used without further purification poly-L-lysine hydrobromide (4–15 kDa, 19–72 residues, monomer: 209 g/mol), poly-L-lysine hydrobromide (1–5 kDa, 5–24 residues, monomer: 209 g/mol). All RNA and DNA oligomers were obtained from Integrated DNA Technologies (Coralville, IA) and are listed in *Tables 1 and 2*. Fluorescently labelled oligomers were ordered with HPLC purification, whilst unlabelled oligomers were desalted and used without further purification. All RNA oligomers were dissolved in RNase-free water and stored at –80 °C.

## Methods
### Preparation of RNA

RNA was ordered from IDT or transcribed in-house from DNA templates. DNA sequences containing a T7 promoter upstream of the RNA sequence of interest were ordered from IDT. The DNA templates were prepared by annealing a complementary oligo to the T7 promoter region by heating equimolar mixture of the oligos at 85°C then cooling on ice. The resulting partially double-stranded DNA was used as DNA template for transcriptions. Large-scale in vitro transcription (IVT) reactions were adopted to produce enough RNA for downstream applications. The transcription reaction volume varied from 400 µL to 4 mL and contained the following: 1 µM partially double-stranded DNA template, 30 mM Tris-HCl pH 7.8, 30 mM $MgCl_2$, 10 mM DTT, 2 mM spermidine, 5 mM of each NTP, 1 U/mL inorganic pyrophosphatase, and 0.5 µM T7 RNA polymerase (purified from a recombinant source in-house). The reaction proceeded for 4–6 hr at 37°C, after which the volume was concentrated by spinning at 15,000 × *g*, 4°C in Amicon ultrafiltration columns (Merck) with 3 kDa molecular weight cut-off

regenerated cellulose filters. Following concentration, the RNA was purified with the Monarch RNA cleanup kit (NEB) following the manufacturer's instructions, eluted twice in water, and quantified on the NanoDrop OneC (Thermo Fisher Scientific). The resulting ribozymes were gel purified using 12% urea PAGE, whereas the substrate was purified using 20% PAGE. The amount of RNA loaded per well was 5–8 µg mm$^{-2}$ of the surface area of the bottom of the well. After PAGE, the gel was wrapped in plastic foil and the band of interest was identified by UV shadowing (254 nm, <30 s) on an autofluorescent background. The gel slice was excised, crushed, weighed, and soaked in 2 µL of 0.3 M sodium acetate (pH 5.2) per mg of gel at 4°C overnight. Following elution, the gel debris was removed using Costar Spin-X columns (Corning Inc, Corning, NY) with 0.45 µM cellulose acetate filters. Next, 20 µg RNA -grade Glycogen (Invitrogen, Waltham, MA) and 1.2 volumes of cold isopropanol were added to the solution. The mixture was cooled for 1 hr at –20°C to promote precipitation, and then centrifuged for an additional hour at 21,000 × *g*, 4°C. The supernatant was removed, and the pellet washed twice with 0.5 volumes of cold 80% ethanol. After washing, the supernatant was completely removed, and the pellet was dried for a few minutes under vacuum and resuspended in ultrapure water. RNA was aliquoted and stored at –80°C.

## Urea polyacrylamide gel electrophoresis (PAGE)

Rotiphorese Gel 40 (19:1) was used to prepare 20% polyacrylamide gel stocks containing 8 M urea and 1× TBE (89 mM Tris, 89 mM boric acid, 2 mM EDTA, pH 8). A 0% gel stock was prepared using the same volumes with water replacing acrylamide. The two stocks were mixed at different ratios to obtain the final desired acrylamide concentration. Polymerisation was initiated by adding 0.01 volumes of 10% APS and 0.001 volumes of TEMED. The gel was cast in an EasyPhor PAGE Maxi Wave (20 cm × 20 cm) (Biozym, Hessisch Oldendorf) and allowed to polymerise at room temperature for >2 hr, then pre-run at a constant power of 20 W for 45 min. The gel thickness was 2 mm and 1 mm for preparative and analytical gels, respectively. The quenched samples were loaded, and the gel was run for 90 min at 20 W constant power in 1× TBE running buffer. When necessary, the gel was removed from the glass plates and stained with SYBR Gold (Invitrogen) nucleic acid staining dye for 10 min in 1× TBE and 1× SYBR Gold, then washed twice for 5 min in de-ionised water to decrease background fluorescence. Images were acquired using and Azure Sapphire RGB gel scanner ($\lambda_{ex}$ = 520 nm for SYBR Gold, 658 nm for Cy5) and analysed with the AzureSpot software (Azure Biosystems, Dublin, CA). The images were background subtracted using the built-in rolling ball function set to a diameter of 1000.

## Poly(L-lysine) titration PAGE

Ribozyme assays were carried out with a 10 µL total reaction volume and the following components: 50 mM Tris-HCl pH 8.6, 10.5 µM ribozyme, 10.5 µM substrate, 10 mM MgCl$_2$, and varying concentrations of poly(L-lysine). Ratios of RNA to (Lys)$_n$ were based on a fixed charge concentration of the RNA (1 mM total monomer charge), calculated with the following equation:

$$Total\ Negative\ Charge = \left(c_{ribozyme} \times l_{ribozyme}\right) + \left(c_{substrate} \times l_{substrate}\right)$$

Positive charge concentrations were calculated based on the lysine hydrobromide monomer repeat molecular weight (209 g/mol) and mass of poly(L-lysine), considering a single positive charge for each residue. Reactions were set up at room temperature by first adding the all the components except the RNA to allow equilibration of poly(L-lysine) in the buffer. The reaction was then started by adding a mixture of ribozyme and substrate and incubated in a thermocycler at 30°C for 2 hr. Reactions were stopped by adding 1 volume of 5 M NaCl, 1 volume of 1.25 M hexametaphosphate (HMP), and 12 volumes of RNA loading buffer containing 10 mM EDTA, 0.05% bromophenol blue, 95% formamide. The resulting samples were briefly vortexed, denatured for 5 min at 85°C, cooled quickly on ice, and centrifuged for 5 min at 2000 × *g* (Color Sprout Plus, Biozym). PAGE analysis proceeded as described above.

## OD measurements

Measurements were performed using a NanoDrop OneC (Thermo Fisher Scientific) by measuring absorbance at 500 nm as a proxy for coacervate formation. The RNAs were pre-reacted at 2× concentration (21 µM each) in 1× buffer (50 mM Tris-HCl pH 8.6, 10 mM MgCl$_2$) for 3 hr at 30°C. Varying concentrations of poly(L-lysine) in 1× buffer were then added to the reacted RNA. For every

poly(L-lysine) concentration tested, 2 µL of pre-reacted RNA was added to 2 µL of poly(L-lysine) and mixed by pipetting 10 times. The resulting solution was incubated for 5 min, after which absorbance was measured. At least three biological replicates were measured for each concentration. From these data, we selected specific peptide:RNA ratios for both peptides for further experimentation (0.75:1 $(Lys)_{19-72}$):RNA and (3:1 $(Lys)_{5-24}$:RNA). The criterion for this selection was the formation of liquid coacervate droplets (determined by turbidity measurements and microscopy) at a peptide:RNA ratio that did not supress ribozyme activity (determined by PAGE analysis of lysine titration). Both concentrations occur shortly before the respective turbidity maxima in the peptide titration turbidity curve.

## Ribozyme kinetics

Time-dependent assays were performed using individual aliquots for each time point to mitigate the effects of coacervate adhesion to the PCR tube walls over time. The reaction mixture contained 50 mM Tris-HCl pH 8.6, 10 mM $MgCl_2$, 1 mM RNA charge concentration (9.5 µM substrate, 1 µM Cy5-tagged substrate, 10.5 µM ribozyme), and either 3 mM $(Lys)_{5-24}$ or 0.75 mM $(Lys)_{19-72}$ charge concentration. Aliquoted reactions were quenched by the addition of 2 µL 5 M NaCl, 2 µL 1.25 M HMP, and 34 µL RNA loading buffer (10 mM ETDA, 0.05% bromophenol blue, 95% formamide) at designated time points (t = 0, 2, 5, 10, 20, 30, 60, 120, 180, and 240 min). Sample preparation and PAGE were performed as described above. The average substrate length and relative band percentages were calculated as follows:

$$average\ substrate\ length\ (\mathrm{nt}) = \sum product\ length \times relative\ abundance$$

$$relative\ abundance = \frac{intensity\ of\ band}{total\ intensity\ of\ lane} \times 100$$

The data were fitted in GraphPad Prism using both a first-order ('one-phase association') and second-order ('two-phase association') kinetic model. First- and second-order kinetics were discriminated between using the extra sum-of-squares $F$ test for nested models. The simpler first-order model was rejected when $p<0.05$.

## Temperature and ratio screen

Reactions were carried out with varying ribozyme to substrate ratios, maintaining a total negative charge of 1 mM. Reactions with component ratios of 1:1, 1:2, and 1:4 contained ribozyme:substrate concentrations 10.5:10.5 µM, 8:16 µM, and 5:20 µM, respectively. Buffer conditions were 50 mM Tris-HCl, pH 8.6, and 10 mM $MgCl_2$. The reactions were incubated for 2 hr at 30, 37, and 45°C, then quenched by the addition of nine volumes of RNA loading buffer (10 mM EDTA, 0.05% bromophenol blue, 95% formamide). They were resolved on an 8% urea PAGE and stained and imaged as described previously.

## RNase R digestion

The reaction products from 10 µL reactions containing 1:1 ribozyme:substrate with either no poly(L-lysine), 0.75:1 $(Lys)_{19-72-15}$ or 3:1 $(Lys)_{5-24}$ and incubated for 3 hr at 45°C were recovered using a 50 µg Monarch RNA cleanup kit (NEB) and eluted in 12 µL ultrapure water. RNase R (Applied Biological Materials, Richmond, Canada) was used to digest a portion of the purified reaction products. The digest reaction mixture (20 µL) contained 1 µg of RNA, 1× RNase R buffer, and 30 U of RNase R, and was incubated for 3.5 hr at 37°C. The digested RNA was recovered with the 10 µg Monarch RNA cleanup kit (NEB), eluted in 6 µL of ultrapure water and mixed with 6 µL of RNA loading buffer. 1.5 µL of the undigested RNA and 5 µL of the digested RNA were resolved on an 8% PAGE, stained, and imaged as described above.

## PEG 8000 and magnesium chloride screens

Reactions were carries out in the standard buffer system (50 mM Tris-HCl pH 8.6 and 10 mM $MgCl_2$), and were supplemented with either PEG 8000 alone, $MgCl_2$ alone, or both in combination. All reactions comprised a 1:1 ribozyme to substrate ratio and final positive charge concentration of 1 mM (10.5 µM final concentration of each RNA with 10% Cy5-tagged substrate). The reactions were set up on ice, started by adding the RNAs and incubated at 30°C for 10 min. After incubation, the reactions containing either PEG 8000 or $MgCl_2$ were quenched with 20 volumes of RNA loading buffer

(10 mM EDTA in formamide). For the conditions that combine PEG 8000 and $MgCl_2$ a positive control with added $(Lys)_n$ was performed. Consequently, these reactions were quenched by the addition of 1 volume of 5 M NaCl, followed by 1 volume of 1.25 M hexametaphosphate and 18 volumes of RNA loading buffer. All samples were heat denatured for 5 min at 85°C and subsequently resolved on an 8% urea PAGE and imaged as described above.

### RNA concentration screen

Reactions were carried out in the standard buffer system (50 mM Tris-HCl pH 8.6 and 10 mM $MgCl_2$) with different concentrations of total RNA but maintaining the same 1:1 ratio of ribozyme to substrate. The concentration of each RNA was either 5.25 µM designated as 0.5×, 10.5 µM designated as 1×, or 21 µM designated as 2×. All reactions had 10% Cy5-tagged substrate included. The reactions were set up on ice followed by a 10 min incubation period at 30°C. The reactions were quenched with different volumes of RNA loading buffer such that the final concentration of labelled species is the same in the quenched sample buffer (half the volume of leading buffer for the 0.5× and twice the volume of loading buffer for the 2×). The 1× reaction was quenched with four volumes of RNA loading buffer. All samples were heat denatured for 5 min at 85°C and subsequently resolved on an 8% urea PAGE and imaged as described above.

### Microscopy

Microscopy was performed on a Leica Thunder inverted widefield microscope equipped with a sCMOS camera Leica DFC9000 GTC using a ×63/NA 1.47 objective. Fluorescence channels were $\lambda_{Ex}$ = 484–496 nm/ $\lambda_{Em}$ = 507–543 nm for FAM, and $\lambda_{Ex}$ 629–645 nm/ $\lambda_{Em}$ 669–741 nm for Cy5. The sample stage was warmed to 30°C. Samples were loaded into clear-bottomed 384-well plates (Greiner µclear, medium binding). Coacervate droplets were formed by combining a solution of RNA with a solution containing poly(L-lysine), buffer and magnesium chloride and mixing using a pipette in a PCR tube. The mixture was incubated on ice for 5 min, then 5 µL was loaded into the well plate. Individual wells were sealed with a drop of silicon oil to prevent evaporation. After loading, plates were immediately incubated at 30°C. Imaging was performed as soon as the coacervate suspension had settled, typically first at 30 min after mixing. A 3 × 3 image grid, centred at the middle of the well, was captured for each sample at various time points. In most cases multiwell plates were passivated using Pluronic F-68 to prevent droplet wetting and adhesion.

### Image processing and analysis

Following data collection, grid tiles were inspected, and out-of-focus images were discarded. Only images that remained in focus for all time points in a given sample were carried forward for analysis. Image masks were produced by segmentation in the brightfield channel using the CellPose algorithm (cyto2 model, average cytoplasm diameter = 30–100 pixels, flow threshold = 0.4, cell probability threshold = 0) (*Stringer et al., 2021*). A rolling ball background was subtracted in all fluorescence channels (radius = 100 pixels). Masks and images were then used to measure the number of particles per image, particle area, and volume corrected fluorescence intensity. Both inter- and intra-particle PCCs were calculated for samples containing two populations of droplets with orthogonal labels. For figure preparation, time-course images were aligned using HyperStackReg (v.5.6, translation transformation) and displayed without background subtraction (*Sharma, 2018*).

## Acknowledgements

We thank Martin Spitaler, Markus Oster, Giovanni Cardone, and the MPIB Imaging Core Facility for providing excellent guidance throughout the project, as well as subsidised equipment access. HM and ES were supported by funding from the Deutsche Forschungsgemeinschaft (DFG, German Research Foundation, Project-ID 364653263-TRR 235). HM gratefully acknowledges support by the European Research Council (ERC) under the Horizon 2020 research and innovation programme (grant agreement ID: 802000, RiboLife). KLV, BG, T-Y DT, and HM were supported by the Volkswagen Foundation with funding from the initiative 'Life? - A Fresh Scientific Approach to the Basic Principles of Life' (grant number: 92772).

## Additional information

### Funding

| Funder | Grant reference number | Author |
|---|---|---|
| Deutsche Forschungsgemeinschaft | 364653263 | Elia Salibi<br>Hannes Mutschler |
| European Research Council | 802000 | Hannes Mutschler |
| Volkswagen Foundation | 92772 | Kristian Kyle Le Vay<br>Basusree Ghosh<br>TY Dora Tang<br>Hannes Mutschler |

 The funders had no role in study design, data collection and interpretation, or the decision to submit the work for publication.

### Author contributions

Kristian Kyle Le Vay, Conceptualization, Resources, Formal analysis, Supervision, Funding acquisition, Investigation, Visualization, Methodology, Writing – original draft, Project administration, Writing – review and editing; Elia Salibi, Formal analysis, Investigation, Visualization, Methodology, Writing – original draft, Writing – review and editing; Basusree Ghosh, Investigation, Methodology, Writing – review and editing; TY Dora Tang, Conceptualization, Resources, Supervision, Funding acquisition, Writing – review and editing; Hannes Mutschler, Conceptualization, Resources, Supervision, Funding acquisition, Methodology, Project administration, Writing – review and editing, Writing – original draft

### Author ORCIDs

Kristian Kyle Le Vay (iD) http://orcid.org/0000-0003-2455-8706
Elia Salibi (iD) http://orcid.org/0009-0003-8201-4237
Hannes Mutschler (iD) http://orcid.org/0000-0001-8005-1657

### Decision letter and Author response

Decision letter https://doi.org/10.7554/eLife.83543.sa1
Author response https://doi.org/10.7554/eLife.83543.sa2

## Additional files

### Supplementary files

• MDAR checklist

### Data availability

All data generated during this study are included in the manuscript and supporting files; source data files have been provided for all figures and figure supplements.

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
