## [Editor Report]

Experimental models of simple cell-like compartments can help us to understand how biology operated early in its history. The authors convincingly show how the properties of coacervate droplets can be influenced by the activity of ribozymes inside them. This important result potentially provides a new route for biologists or chemists to establish cell mimics that support the evolution of biomolecules within.

---

## [Decision Letter]

**Decision letter after peer review:**

Thank you for submitting your article "Ribozyme-phenotype coupling in peptide-based coacervate protocells" for consideration by *eLife*. Your article has been reviewed by 3 peer reviewers, and the evaluation has been overseen by Timothy Nilsen as the Reviewing Editor and James Manley as the Senior Editor. The reviewers have opted to remain anonymous.

The reviewers have discussed their reviews with one another, and the Reviewing Editor has drafted this to help you prepare a revised submission. (Please note that point 3 should have read: Paragraphs 3-5 of reviewer 2's author recommendations.)

Essential revisions:

The reviewers were generally quite positive about the work. They have made a number of suggestions that may be useful for future investigation in this area. Of these, some of these are essential as detailed below. We are not asking for any additional experiments, but please discuss the points raised by the reviewers.

1. Clarification of the title.

2. The results shown in figure 1b, 1d are discordant and this needs to be remedied by either choosing different images or (not as good) explaining in the text why these panels seem to be discordant.

3. Paragraphs 3-5 of reviewer 2's author recommendations.

*Reviewer #1 (Recommendations for the authors):*

1. As the products of the ligation reaction by the EL R3C ladder ribozyme are mixtures of RNAs with different lengths, the exact requirements (e.g., average/maximum length of the ligated RNAs + double strand formation) for the phenotypic changes are not clear. Is S2 or S3 enough? Are longer products (e.g., S6-S17) more important? Do the elongated RNAs need to remain hybridized with the ribozymes? Droplet formation experiments with S2 or S3 (synthesized chemically or by T7 RNAP) in the presence and absence of the inactive hybridizing ribozyme would clarify these points.

2. It is unclear if the droplet property changes observed in this study are advantageous for the contained ribozyme. Is the EL R3C ladder ribozyme still active in the more viscous/solid droplets containing the longer RNAs? A ribozyme activity test started from a substrate long enough to form more viscous/solid droplets might answer this question.

*Reviewer #2 (Recommendations for the authors):*

The public review reflects my general view of the paper: it is an impressive achievement to get a ribozyme to control coacervate properties. The authors propose to exploit this by using the droplets to implement selection. There are still questions about the system's suitability for such a task. Readers would benefit from a clearer path forward from the authors, including an experimental assessment of a couple of important points.

A direct test of coacervate suitability for such a task would be to carry out a model in vitro selection of ligases based on their activity and influence on coacervate properties, though the system may not yet be at a stage where this is possible. In the meantime some clarity on the following points would help assess the prospects of this:

1) Coacervate enhancement of ribozyme activity: The results in Figure 1d and Figure 1b seem discordant. At the 2 h timepoint in Figure 1d, the average product length seems much lower without Lys than with – but the equivalent lanes look the same in 1b with (as far as I can tell) identical conditions. Visual inspection of the gel in 1b also suggests a much higher average product length than calculated in 1d (though perhaps this is a mirage due to staining intensity) and seen in the supporting data. If I'm missing something in my interpretation, perhaps the authors could more clearly explain the differences. This is important for readers to grasp as it underpins the claim that coacervate formation enhances ribozyme activity. I'd recommend showing a gel from the supporting figure clearly showing a condition where ligation is better with Lys. It may also be helpful to quantify the average product length in SYBR Gold stained gels (correcting for the proportionality of RNA length to signal).

2) Ribozyme activity: the observed effects of ribozyme activity on droplet properties occur when ~1/4 of the coacervate comprises ribozyme. For selection, there would be much less active ribozyme present. Can small amounts of ribozyme still influence droplet behavior when at lower concentrations (e.g. a mix of active and inactive ribozymes)?

3) Droplet fusion: There is convincing resistance of droplets to fusion when both contain active ribozyme. But in a selection setting, where a small fraction of droplets have active ribozyme and the majority do not, the relevant question is whether the droplets with inactive ribozyme will fuse with droplets with active ribozyme, i.e. does ribozyme activity in a droplet make it resistant to fusion or remove its tendency to fuse?

*Reviewer #3 (Recommendations for the authors):*

1 – The sentence "The increase in rate observed here is most likely due to the high concentration of ribozyme and substrate that occurs in coacervates directly formed from catalytic RNA and peptides (Ref. 15)" addresses an interesting point: How do we know that the peptide-mediated increase in ribozyme activity is due to coacervate formation as opposed to, for example, an association of freely diffusing peptides with freely diffusing ribozymes? Given that this is not the central point of the paper I would not ask for (probably difficult) experiments but for 1-2 more sentences based on reference 15 that increase the reader's understanding of how the peptide likely mediates through coacervates the increased ribozyme activity.

2 – The manuscript is of high interest in the field of origins of life, and needs to discuss in more detail the important implications of coacervates as model systems for early life forms, especially with regard to the effects of ribozyme catalysis on the material exchange between coacervate droplets. Specifically, the coupling of genotype and phenotype, the danger from molecular parasites, and the inflow/outflow of metabolites are likely affected by these changes, and this should be discussed based on the rich literature. I suggest inserting an additional paragraph into the discussion or expanding the last paragraph on these three specific points.

3 – While the longer Lys(19-72) peptides do not support ribozyme catalysis (see the first part of the results) they are used as a model system for dye exchange (Figure 3). Please state in the corresponding text the caveat that these longer peptide systems are not model systems for protocells but serve to test the effects of peptide length on dye exchange. Please also discuss whether the less pronounced effect on the exchange rates in coacervates that support ribozyme catalysis would be sufficient for evolutionary importance with respect to protocell function (see (2) above).

4 – The discussion mentions that the Lys(5-24) peptides are actually not 5-24 amino acids long but 3 to 9 amino acids in length. To avoid confusing the reader, please correct the length throughout the manuscript, and mention in the materials and methods that "commercially available (Lys)n deviates from the manufacturer's stated size range, with the (Lys)5-24 used here being predominantly comprised of oligomers between n = 3 to n = 9 in length15.".

---

## [Author Response]

Essential revisions:The reviewers were generally quite positive about the work. They have made a number of suggestions that may be useful for future investigation in this area. Of these, some of these are essential as detailed below. We are not asking for any additional experiments, but please discuss the points raised by the reviewers.1. Clarification of the title.

We agree that the previous title was somewhat ambiguous and have therefore incorporated the suggestions of reviewers 1 and 3 to retitle the paper:

Ribozyme activity modulates the physical properties of RNA-peptide coacervates

2. The results shown in figure 1b, 1d are discordant and this needs to be remedied by either choosing different images or (not as good) explaining in the text why these panels seem to be discordant.

The combination of endpoint and kinetic data is potentially confusing, and the effect of phase separation is not obvious from the SYBR Gold stained urea-PAGE gels of endpoint data. However, the use of a stained gel is necessary to visualise all of the components of the system. We have therefore attempted to clarify the data with the following measures:

Lane profiles for relevant conditions (solution and 0.75:1 pL:RNA for Figure 1b, solution and 1.5:1 pL:RNA for Figure 1 – supplement 2) have been incorporated into the relevant figures. These profiles illustrate that the intensities of the ligated products are higher in the phase separated system, an effect which is not easily perceived in the SYBR Gold stained gel image.Images of the source urea-PAGE gels for the kinetic experiments (Figure 1e) have been converted into a new figure supplement (Figure 1 – supplement 3), which clearly shows the difference in rate and activity when monitoring the extension of a fluorescently-tagged substrate strand.We have added some brief explanation to the text to account for the discordance and to draw attention to the newly added features and figures.

We hope that these additions are sufficient to remedy the issue and discuss the issues further in our responses to reviewers 1 and 2.

3. Paragraphs 3-5 of reviewer 2's author recommendations.

Please see author recommendations for detailed responses and revisions.

Reviewer #1 (Recommendations for the authors):1. As the products of the ligation reaction by the EL R3C ladder ribozyme are mixtures of RNAs with different lengths, the exact requirements (e.g., average/maximum length of the ligated RNAs + double strand formation) for the phenotypic changes are not clear. Is S2 or S3 enough? Are longer products (e.g., S6-S17) more important? Do the elongated RNAs need to remain hybridized with the ribozymes? Droplet formation experiments with S2 or S3 (synthesized chemically or by T7 RNAP) in the presence and absence of the inactive hybridizing ribozyme would clarify these points.

We thank the reviewer for the positive assessment of the work and helpful suggestions. Indeed, the requirements for altering material properties in these experiments are not precisely known. Follow-up studies performed in the manner suggested by the reviewer could elucidate these questions, by determining the physical properties of droplets formed from pre-reacted substrate concatemers with defined lengths or in the presence of a non- or weakly-hybridising ribozyme.

In designing our experiments, we relied on previous works (e.g. Spruijt *et al.* 2010) that investigated the effect of polymer length on coacervate physical properties to roughly determine initial and final RNA lengths that were likely to result in an observable change.

However, the exact conditions for achieving phenotypic change in a given system are highly specific and dependent on a broad parameter space that encompasses polymer length, RNA sequence, hybridisation state, overall polymer concentration, buffer pH, salt concentration and Mg^2+^ concentration amongst other factors. As a result, determining the exact requirements would be valuable for the further development of our model system, but would not provide general rules applicable to RNA-peptide coacervates.

2. It is unclear if the droplet property changes observed in this study are advantageous for the contained ribozyme. Is the EL R3C ladder ribozyme still active in the more viscous/solid droplets containing the longer RNAs? A ribozyme activity test started from a substrate long enough to form more viscous/solid droplets might answer this question.

The current ribozyme and substrate design did not allow us to directly address this question, as the ribozyme catalyses a pseudo-irreversible reaction that reaches an endpoint over the course of the experiment. As suggested, increasing the initial substrate length with an unstructured spacer region would be a good starting point to investigate this question.

In our previous work (Le Vay *et al.* 2021, https://doi.org/10.1002/anie.202109267), we reported that highly viscous or gel-like phases can occur at specific ratios in mixtures of a fragmented hairpin ribozyme and poly(L-lysine), albeit at a much lower overall polymer length. The formation of these phases was associated with increased activity, and both ribozyme activity and strand exchange were maintained, suggesting that high viscosity is not a barrier to ribozyme function in specific cases.

Reviewer #2 (Recommendations for the authors):The public review reflects my general view of the paper: it is an impressive achievement to get a ribozyme to control coacervate properties. The authors propose to exploit this by using the droplets to implement selection. There are still questions about the system's suitability for such a task. Readers would benefit from a clearer path forward from the authors, including an experimental assessment of a couple of important points.A direct test of coacervate suitability for such a task would be to carry out a model in vitro selection of ligases based on their activity and influence on coacervate properties, though the system may not yet be at a stage where this is possible. In the meantime some clarity on the following points would help assess the prospects of this:1) Coacervate enhancement of ribozyme activity: The results in Figure 1d and Figure 1b seem discordant. At the 2 h timepoint in Figure 1d, the average product length seems much lower without Lys than with – but the equivalent lanes look the same in 1b with (as far as I can tell) identical conditions. Visual inspection of the gel in 1b also suggests a much higher average product length than calculated in 1d (though perhaps this is a mirage due to staining intensity) and seen in the supporting data. If I'm missing something in my interpretation, perhaps the authors could more clearly explain the differences. This is important for readers to grasp as it underpins the claim that coacervate formation enhances ribozyme activity. I'd recommend showing a gel from the supporting figure clearly showing a condition where ligation is better with Lys. It may also be helpful to quantify the average product length in SYBR Gold stained gels (correcting for the proportionality of RNA length to signal).

We thank the reviewer for the positive assessment of this work and for highlighting several possible improvements.

In response to point 1, we direct the reviewer to our response to Reviewer 1, and to the editorial team in the section “essential revisions”. In brief, the discordance stems from the use of stained gel images to illustrate the overall reaction at its endpoint in Figure 2b, but the use of a fluorescently 5’-labelled substrate strand to determine kinetics in 2d. The use of stained gels allows us to visualise all species, including the ribozyme, substrate and both linear and circular products. However, monitoring the reaction in this way complicates the determination of average substrate length due to problems of band overlap (e.g. ribozyme and S2 product) and the presence of large but low abundance circular products of unknown length, and makes visual assessment of differences in average substrate length at the endpoint of the reaction difficult.

We feel that it is important to present a stained gel which displays all reaction components and products, but have aimed to clarify the data by adding lane profiles for certain conditions, which the greater intensity of linear product bands in the presence of pL, adding a new figure supplement based on the fluorescently labelled gels from the source data of the kinetics experiment, and adding some clarification to the text.

2) Ribozyme activity: the observed effects of ribozyme activity on droplet properties occur when ~1/4 of the coacervate comprises ribozyme. For selection, there would be much less active ribozyme present. Can small amounts of ribozyme still influence droplet behavior when at lower concentrations (e.g. a mix of active and inactive ribozymes)?

This is a valid point that we feel warrants further discussion here and in the manuscript. In a more realistic scenario, it is expected that the concentrations of both functional ribozyme and compatible substrate would be much lower, and that the remaining material would contain non-functional (or at least non-ligating) RNA strands and perhaps other charged polymers.

It is therefore necessary to improve the catalytic properties of the ribozyme in order to move towards a more realistic system. In Figure 1 – supplement 1, we show that increasing the amount of substrate relative to the ribozyme actually leads to a reduction in ribozyme activity. We therefore anticipate that optimising the ribozyme for multi-turnover reaction might be required to achieve significant phenotypic change in a system with small amounts of ribozyme but abundant substrate. In our previous work, we demonstrated that the Hairpin ribozyme is capable of strand exchange and therefore multiple substrate ligations in RNA-peptide condensates, perhaps due to the short length of the substrate binding strand. By altering the substrate binding arms of the R3C ribozyme, it might be possible to engineer similar behaviour, perhaps at the cost of reaction rate.

In more complex mixtures with a greater variety of RNA strands and therefore a less well-defined substrate pool, a ribozyme with less stringent substrate sequence requirements might be better suited to alter material properties. The *sunY* ribozyme, which requires only a 5’-G for substrate ligation, would be an interesting candidate for this.

We have modified the final paragraph of the discussion to account for some of these shortcomings.

3) Droplet fusion: There is convincing resistance of droplets to fusion when both contain active ribozyme. But in a selection setting, where a small fraction of droplets have active ribozyme and the majority do not, the relevant question is whether the droplets with inactive ribozyme will fuse with droplets with active ribozyme, i.e. does ribozyme activity in a droplet make it resistant to fusion or remove its tendency to fuse?

This is a highly relevant but experimentally challenging question, which we are currently attempting to address as part of our ongoing projects. The orthogonally labelled substrates transfer between droplet populations by both fusion and diffusion, and as a result all droplets contain measurable amounts of both fluorophores shortly after mixing. This means we are currently unable to investigate the behaviour of each droplet population independently to determine if e.g. an active population resists fusion when mixed with an inactive population. Our preliminary results suggest that in mixtures of active and inactive droplets, the labelled RNA from the active population remains highly localised, whilst the labelled RNA from the inactive population diffuses evenly throughout both populations. Further work is required to investigate fusion in these systems, in particular to establish methods of orthogonally measuring droplet size in the two populations.

Reviewer #3 (Recommendations for the authors):1 – The sentence "The increase in rate observed here is most likely due to the high concentration of ribozyme and substrate that occurs in coacervates directly formed from catalytic RNA and peptides (Ref. 15)" addresses an interesting point: How do we know that the peptide-mediated increase in ribozyme activity is due to coacervate formation as opposed to, for example, an association of freely diffusing peptides with freely diffusing ribozymes? Given that this is not the central point of the paper I would not ask for (probably difficult) experiments but for 1-2 more sentences based on reference 15 that increase the reader's understanding of how the peptide likely mediates through coacervates the increased ribozyme activity.

We would like to thank the reviewer for this point, and agree that the claim was not fully supported. In response to this and a similar point raised by Reviewer 1, we have performed additional experiments investigating the activity of the ribozyme at varying PEG, Mg^2+^ and RNA concentrations. These experiments reveal that both high RNA and Mg^2+^ concentrations are required for enhanced activity. We direct the reviewer to the earlier response to Reviewer 1, and to the new data presented in Figure 1 – supplement 4.

2 – The manuscript is of high interest in the field of origins of life, and needs to discuss in more detail the important implications of coacervates as model systems for early life forms, especially with regard to the effects of ribozyme catalysis on the material exchange between coacervate droplets. Specifically, the coupling of genotype and phenotype, the danger from molecular parasites, and the inflow/outflow of metabolites are likely affected by these changes, and this should be discussed based on the rich literature. I suggest inserting an additional paragraph into the discussion or expanding the last paragraph on these three specific points.

We agree that the ribozyme induced changes to coacervate properties may impact factors relevant to early compartmentalised replicators such as parasite resistance. However, as the ribozyme system described in this work is not replicative, we would like to limit the discussion of these aspects to avoid lengthening the manuscript further. We have included a sentence in the discussion highlighting these areas as interesting avenues for further study.

3 – While the longer Lys(19-72) peptides do not support ribozyme catalysis (see the first part of the results) they are used as a model system for dye exchange (Figure 3). Please state in the corresponding text the caveat that these longer peptide systems are not model systems for protocells but serve to test the effects of peptide length on dye exchange. Please also discuss whether the less pronounced effect on the exchange rates in coacervates that support ribozyme catalysis would be sufficient for evolutionary importance with respect to protocell function (see (2) above).

We thank the reviewer for bringing this miscommunication to light. The longer lysine polymers do indeed inhibit ribozyme activity, but only at ratios when the lysine oligopeptides are in excess over RNA i.e. (Lys)_19-72_: RNA > 1:1, as determined by the lysine titration PAGE in Figure 1b. In subsequent experiments, we selected a (Lys)_19-72_: RNA ratio of 0.75:1 at which the compartmentalised ribozymes are highly active and coacervates form in liquid state. We believe that under these conditions the droplets containing longer peptides are an interesting model system. In addition, all experiments are also performed with droplets formed from the shorter peptide and the difference in behaviour between these systems is discussed. This point has been clarified in the second paragraph of the Results section.

4 – The discussion mentions that the Lys(5-24) peptides are actually not 5-24 amino acids long but 3 to 9 amino acids in length. To avoid confusing the reader, please correct the length throughout the manuscript, and mention in the materials and methods that "commercially available (Lys)n deviates from the manufacturer's stated size range, with the (Lys)5-24 used here being predominantly comprised of oligomers between n = 3 to n = 9 in length15.".

We thank the reviewer for bringing this miscommunication to light. The longer lysine polymers do indeed inhibit ribozyme activity, but only at ratios when the lysine oligopeptides are in excess over RNA i.e. (Lys)_19-72_: RNA > 1:1, as determined by the lysine titration PAGE in Figure 1b. In subsequent experiments, we selected a (Lys)_19-72_: RNA ratio of 0.75:1 at which the compartmentalised ribozymes are highly active and coacervates form in liquid state. We believe that under these conditions the droplets containing longer peptides are an interesting model system. In addition, all experiments are also performed with droplets formed from the shorter peptide and the difference in behaviour between these systems is discussed. This point has been clarified in the second paragraph of the Results section.